# Unleashing the Potential of EIL Transcription Factors in Enhancing Sweet Orange Resistance to Bacterial Pathologies: Genome-Wide Identification and Expression Profiling

**DOI:** 10.3390/ijms241612644

**Published:** 2023-08-10

**Authors:** Yajun Su, Suming Dai, Na Li, Alessandra Gentile, Cong He, Jing Xu, Kangle Duan, Xue Wang, Bing Wang, Dazhi Li

**Affiliations:** 1National Citrus Improvement Center, Hunan Agricultural University (Changsha Branch), Changsha 410128, China; 2College of Horticulture, Hunan Agricultural University, Changsha 410128, China; 3College of Plant Protection, Hunan Agricultural University, Changsha 410128, Chinaxuewang20220908@163.com (X.W.); 4Department of Agriculture and Food Science, University of Catania, 95123 Catania, Italy; gentilea@unict.it

**Keywords:** *Citrus sinensis*, EIL family, abiotic stress, citrus canker, transcription factor

## Abstract

The ETHYLENE INSENSITIVE3-LIKE (EIL) family is one of the most important transcription factor (TF) families in plants and is involved in diverse plant physiological and biochemical processes. In this study, ten EIL transcription factors (CsEILs) in sweet orange were systematically characterized via whole-genome analysis. The *CsEIL* genes were unevenly distributed across the four sweet orange chromosomes. Putative *cis*-acting regulatory elements (CREs) associated with *CsEIL* were found to be involved in plant development, as well as responses to biotic and abiotic stress. Notably, quantitative reverse transcription polymerase chain reaction (qRT-PCR) revealed that *CsEIL* genes were widely expressed in different organs of sweet orange and responded to both high and low temperature, NaCl treatment, and to ethylene-dependent induction of transcription, while eight additionally responded to *Xanthomonas citri* pv. *Citri* (Xcc) infection, which causes citrus canker. Among these, *CsEIL2*, *CsEIL5* and *CsEIL10* showed pronounced upregulation. Moreover, nine genes exhibited differential expression in response to *Candidatus* Liberibacter asiaticus (*C*Las) infection, which causes Citrus Huanglongbing (HLB). The genome-wide characterization and expression profile analysis of *CsEIL* genes provide insights into the potential functions of the *CsEIL* family in disease resistance.

## 1. Introduction

Transcription factors (TFs) are DNA-binding proteins that specifically interact with *cis*-elements in the promoter regions of genes. Typically, the expression of pathogen-responsive genes is primarily controlled by the activity of specific TFs that bind to specific regulatory sites upstream of the constituent genes through direct physical interactions or conjugation with other proteins in the regulatory network [1]. In plants, more than 50 TF families have been identified by sequence analysis [2]. Members of the NAC (NAM-ATAF1/2-CUC2), WRKY (WRKYGQK), C2H2 (Cys2/His2), AP2/EREBP (APETALA2/ethylene-responsive element binding proteins), and MYB (v-myb avian myeloblastosis viral) families have been well-characterized in *Arabidopsis thaliana* and rice (*Oryza sativa*) [3,4,5]. Moreover, there is increasing evidence that ETHYLENE INSENSITIVE3-LIKE (*EIL*) family genes are involved in defense responses to pathogenic infection and environmental stress [6].

Ethylene (ET) is a plant hormone that plays a critical role in regulating plant growth and development, as well as the responses to various environmental stresses [7,8,9]. The EIL is a plant-specific protein that is essential in the ET signaling pathway and regulates the ET-mediated transcriptional cascade by binding to DNA [10]. Research into EIL began in 1993 with the isolation of ETHYLENE-INSENSITIVE3 (*EIN3*) mutants in Arabidopsis and the cloning of EIN3 and its homologs, EIN3-Like1 (EIL1) and EIL2 [11]. In higher plants, EIL comprises a family of small TFs with different numbers of members in different species. For example, six, nine, and seven members have been identified in Arabidopsis, rice, and poplar, respectively [12,13,14].

The EIL family of TFs is widely expressed in higher plants, with significant variations in their levels and patterns [15]. For instance, in *Dianthus caryophyllus*, the *DCEIL1* gene is expressed in multiple tissues, with higher levels in the stems and ovaries. Interestingly, as flowers age, *DCEIL1* transcription slightly increases in stems and ovaries but decreases in the petals [16], suggesting its crucial role in flower development and senescence. Similarly, *NtEIL* genes in *Nicotiana tabacum* are differentially enriched in various organs, with *NtEIL3/4* exhibiting lower levels while *NtEIL1* and *NtEIL2* displaying higher levels in the roots [17]. Investigating the *VR-EIL* gene expression in *Vigna radiata* revealed the localized upregulation of *VR-EIL1* and *VR-EIL2* in the apical hooks, hypocotyls, and roots of young seedlings. Notably, *VR-EIL1* is primarily expressed in yellowing seedlings, whereas *VR-EIL1* and *VR-EIL2* show similar expression levels in green plants [18]. This suggests distinct functions and regulatory mechanisms of *VR-EIL1* and *VR-EIL2* during mung bean development and stress responses. Furthermore, among the ten *EIL* genes in pears of the Rosaceae family, several genes display tissue-specific expression patterns. These findings underscore the diverse functions and regulatory roles of the different *EIL* genes in the growth, development, and specialization of pears.

The EIL protein family participates in various pathways that mediate diverse plant responses. In plant innate immunity, EIL transcription factors directly control the expression of the receptor kinase FLS2, exerting a direct influence on the regulation of innate immune receptors [19]. In Arabidopsis, ET and jasmonic acid induce *EIL* to synergistically counteract necrotrophic fungal infections [20]. In addition, *EIL* negatively regulates the expression of salicylic acid, thereby inhibiting plant immune responses against biotrophic pathogens [21]. The functional loss of *EIN2*, *EIN3*, and *EIL1* enhances cell death and defense responses mediated by *RPW8.1*. Conversely, the overexpression of *EIN3* significantly diminishes *RPW8.1*-mediated cell death and disease resistance [22]. Overall, these findings regarding the molecular mechanisms regulated by EIL family members shed light on promising prospects for improving disease resistance in important crops.

Despite extensive research on *EIL* genes in annual plants, their roles in perennial woody plants remain largely unknown. Citrus is among the world’s top fruit categories and is subjected to several biotic and abiotic stresses, such as diseases, insects, drought, salinity, and extreme temperatures, during growth and development [23]. Among these, citrus canker caused by *Xanthomonas citri* pv. *Citri* (*Xcc*) and Citrus Huanglongbing (HLB) caused by *Candidatus* Liberibacter asiaticus (*C*Las) are the most destructive biotic stresses affecting citrus crops on a global scale, resulting in substantial economic repercussions for the citrus industry [24,25]. Here, ten sweet orange EIL TFs were systematically identified through genome-wide analysis, followed by analysis of their chromosomal location, gene duplication, structural features, conserved structural domains, motifs, and *cis*-regulatory elements (CRE). The expression patterns of these *CsEIL* genes in response to various biotic and abiotic stressors were also examined. The findings provided a foundation for further exploration of the functional and mechanistic characteristics of *EIL* genes in sweet orange.

## 2. Results

### 2.1. Identification and Phylogenetic Analysis of EIL Family Proteins in Sweet Orange

Ten proteins containing EIN3 structural domains were identified in sweet oranges. The open reading frame lengths of the *EIL* transcription factors in sweet orange ranged from 1126 to 1866 bp, with theoretical molecular weights ranging from 43 to 70 kDa and isoelectric points ranging from 5.26 to 9.05 (Appendix A). Predicted subcellular localization analysis revealed that all members were localized in the nucleus.

To analyze the evolutionary diversity of the EIL proteins, a comprehensive maximum likelihood phylogenetic tree was constructed, which included five species: *Citrus sinensis*, rice, *Arabidopsis*, poplar (*Populus trichocarpa*), and apple (*Malus domestica*) (Appendix A). Phylogenetic clustering analysis showed that the EIL proteins could be divided into three subgroups, termed A, B, and C, according to their motifs (Figure 1). Notably, Group C accounted for two-thirds of the total proteins, while Groups A and C each contained one sweet orange protein, namely, CsEIL10 and CsEIL3, respectively. Interestingly, Group B and Group C presented distinct EIL proteins across all five species, suggesting potential correlations and similarities in their evolutionary history. Additionally, within each group, most EIL proteins shared similar physicochemical characteristics, as demonstrated in Appendix A.

### 2.2. Chromosome Location and Duplication Analysis of CsEIL Genes

A chromosomal localization analysis was conducted to determine the chromosomal distribution of *CsEIL* genes, revealing that ten *CsEIL* genes were mainly mapped to chromosomes 2 and 3. The density of *CsEIL* genes varied for each chromosome, with chromosome 3 having the largest number of *CsEIL* genes (five), whereas chromosomes 1 and chromosomes Un had only two genes (*CsEIL3* and *CsEIL10*) (Appendix A).

To investigate the genetic differentiation and gene duplication in the sweet orange EIL family, a collinearity analysis was performed using One-Step MCScan. No homologous gene pairs were found in sweet orange. Further analysis was conducted on sweet orange, Arabidopsis, poplar, and apple, which returned three homologous gene pairs between sweet orange and Arabidopsis, six between sweet orange and poplar, and three between sweet orange and apple (Figure 2, Appendix A).

### 2.3. Structures and Conserved Motifs of CsEIL Genes

The gene structure (exons/introns) and the conserved structural domains/motifs were analyzed to investigate the potential correlation between gene structure and the evolutionary process of *CsEIL* genes. The results showed that all *CsEIL* genes contained either a 5′-UTR or a 3′-UTR, except for *CsEIL9* and *CsEIL10*. Apart from the common EIN3 structural domain, an additional EIN3-superfamily structural domain was present in subfamilies B and C (Figure 3C). Ten conserved motifs were detected in the *CsEIL* genes, designated as motifs 1 to 10, with the number of conserved motifs varying from 4 to 10 among the *CsEIL* genes (Figure 3D). These findings suggest that *CsEIL* genes have undergone structural changes during their evolutionary history, and the presence of conserved motifs indicates that these changes have been constrained to maintain functional relevance.

### 2.4. Prediction of Promoter Cis-Regulatory Elements of CsEIL Genes

The potential functions of *CsEIL* genes in response to various stress conditions and phytohormones were further explored by predicting their promoter CREs. The CREs are located upstream of the promoter region and are essential in regulating gene transcription.

The start codons of *CsEIL* genes were identified, and abiotic stress responses and phytohormone-related elements were selected for further analysis. Figure 4A depicts the differences in the number, location, and type of CREs detected in the promoters of the different *CsEIL* genes. The presence of two or three tandem CREs was noted, several of which overlapped with other CREs. The findings of the subsequent analysis of such CREs involved in different types of stress and phytohormones are presented in Figure 4B. Most *CsEIL* genes contained all types of CREs, suggesting that *CsEIL* genes may have diverse roles in the response to different stresses and phytohormones in the sweet orange.

### 2.5. Gene Expression Pattern Analysis of CsEIL Genes

#### 2.5.1. Characterization of Tissue Expression of CsEIL Genes in Sweet Orange

qRT-PCR was performed to examine the expression profiles of the ten *CsEIL* genes in different tissues (stem, leaf, and fruit) of the sweet orange. *CsEIL1/2/4/6/7/8* displayed significant expression level differences among all tissues, with expression levels in leaves being higher than in stems (Figure 5). These results suggest that *CsEIL* genes may contribute in a distinctive fashion to various biological processes, and are likely to be involved in sweet orange development.

#### 2.5.2. Expression Analysis of CsEIL in Response to Ethylene Treatment

The expression of *CsEIL* genes was evaluated after treatment with the phytohormone ET. Our results demonstrated that the relative expression levels of certain *CsEIL* genes were upregulated at 12 h post treatment (hpt) and 24 hpt. Specifically, *CsEIL2/5/6/8/9/10* all showed increased expression at 12 hpt, with this decreasing at the 24 hpt mark. In contrast, *CsEIL1* was continuously activated by ET and peaked in transcription at 24 hpt. Relative expression of *CsEIL3* was also significantly upregulated at 24 hpt. Conversely, *CsEIL4* was significantly downregulated at 24 hpt (Figure 6). Taken together, these findings indicate that *CsEIL* genes are sensitive to ET treatment and are potentially important in the response to ET signaling in sweet orange.

#### 2.5.3. Expression Profile of CsEIL Genes in Response to *Xanthomonas citri* and *Candidatus Liberibacter asiaticus*

In the expression profile of *CsEIL* genes in sweet orange following *C*Las infection, it was observed that all genes were downregulated except for CsEIL9, which showed a significant upregulation (Appendix A). When compared to the control, *CsEIL1/2/4/5/6/7/8/10* exhibited significant downregulation, with *CsEIL5/8* demonstrating the most pronounced decrease. *CsEIL3*, on the other hand, did not show significant changes compared to the control.

The expression profiles of *CsEIL* genes in sweet orange plants inoculated at different time points with the bacterial pathogen *Xcc*, which causes citrus canker disease, were examined, and the results are shown in Figure 7. Several *CsEIL* genes, such as *CsEIL2/3/4/5/8/10*, exhibited significant upregulation and reached peak expression levels at 2 days post inoculation (dpi), 4 dpi, 6 dpi, or 8 dpi. In contrast, a set of *CsEIL* genes, including *CsEIL1/7/9*, were significantly downregulated following inoculation. Overall, CsEIL proteins play diverse roles in plant defense mechanisms.

#### 2.5.4. Expression Analysis of CsEIL under Different Abiotic Stresses

The expression of *CsEIL* genes under different abiotic stress conditions was analyzed using qRT-PCR. As shown in Appendix A, most *CsEIL* genes responded to different abiotic stresses. Of these, *CsEIL5/9/10* were induced in response to salt and high-temperature stress, with significant upregulation of *CsEIL9/10* observed at 48 hpt under salt stress. Interestingly, certain *CsEIL* genes exhibited contrasting expression patterns under different stress conditions. For instance, *CsEIL1/2/3/4/6/7/8/9/10* displayed significant upregulation following high-temperature stress, but they were significantly downregulated after low-temperature stress. To visualize the expression patterns of *CsEIL* genes after abiotic stress treatments, box-and-whisker plots were presented in Figure 8, containing median values (lines) and interquartile spacings (box ends). As shown in this figure, *CsEIL* genes were markedly upregulated after high-temperature stress and oppositely downregulated following low-temperature and salt stress.

## 3. Discussion

The EIL transcription factor family plays crucial roles in the regulation of several physiological and biochemical processes in plants [26]. However, the functions of *EIL* genes in the immunity of perennial woody plants have not been sufficiently examined. In this study, we conducted a genome-wide analysis of the sweet orange *EIL* gene family to investigate the potential function and expression diversity of *EIL* family members in response to abiotic stress, hormone treatments, and pathogen infection. The number of EIL family members varies significantly among species, with 10 having been identified in the entire genome of sweet orange, 6 in Arabidopsis [27], 16 in apple [14], and 6 in rice [13]. This variation in the number of genes may reflect differences in genome size and ploidy levels among species [28]. The predicted relative molecular weights and isoelectric points of CsEIL proteins suggest that they have distinct physicochemical properties, which may be linked to their diverse roles in plant growth and development, as well as in the response to environmental stress.

Phylogenetic analysis of *EIL* genes revealed three subgroups, similar to the classification of other plant species such as soybean (*Glycine max*) [29] and small black poplar (*Populus* × *xiaohei*) [30]. The high similarity in terms of physicochemical properties among EIL proteins in the same subgroup may perhaps allude to similar conserved functions. Moreover, protein sequences may have evolved to acquire specific functions in response to different selective pressures. More comprehensive functional studies are required to determine the specific roles of each CsEIL protein in sweet orange growth, development, and responses to biotic and abiotic stresses.

In higher plants, *EIL* genes are expressed in a range of tissues; however, their expression levels and patterns are organ specific. For instance, *PbEIL* genes are reportedly expressed in diverse plant tissues, such as roots, stems, leaves, and fruits. Among these, *PbEIL5*, *PbEIL6*, and *PbEIL10* display the highest abundance in leaves, whereas *PbEIL7* exhibits relatively high levels of expression in the roots. Similarly, in tomatoes, *LeEIL1*, *LeEIL2*, and *LeEIL3* are highly enriched in the roots and stems, whereas *LeEIL4* shows its highest expression in the fruits. These patterns suggest a distinct tissue-specific regulation of *EIL* gene transcription [31,32]. Here, it was observed that all ten *CsEIL* genes were expressed in the stems, leaves, and fruits of the plant, and their expression levels varied significantly between different organs. Specifically, *CsEIL1/2/3/4/5/6/7/8* exhibited significantly higher relative expression levels in leaves, indicating their potential involvement in leaf growth and development.

The *EIL* gene has been shown to not only participate in plant development, but also to act as a hub, integrating ET with other signals involved in plant hormone regulation and the stress response [33]. Analysis of the promoter and the expression of the *CsEIL* genes in response to different abiotic stresses revealed the potential functions of the *CsEIL* gene family in the sweet orange. Specifically, promoter analysis highlighted the presence of several CRE types within the *CsEIL* promoter, suggesting that CREs may be involved in different physiological processes and the response to different types of stresses and plant hormones. Expression analysis further supported this view, indicating that *CsEIL* genes respond to various non-biological stresses, with certain genes being induced, whereas others are transcriptionally suppressed by different types of stress. Interestingly, a set of *CsEIL* genes showed opposite expression patterns under different types of stress, highlighting the complex underlying regulatory mechanisms that control *CsEIL* gene expression. Moreover, the observation that different *CsEIL* genes responded in a distinct manner to different non-biological stresses is evidence of the potential functional specialization of members within this gene family.

Certain abiotic and biotic environmental stressors can have significant adverse effects on citrus crops, leading to substantial decreases in yield and quality [34]. In this regard, the *EIL* genes could provide valuable insight, as they play important roles in plant immunity [35,36]. Recently, it was shown that in an *ein3/eil1* double mutant of Arabidopsis, ET could activate the SA signaling pathway together with many transcription factors, including WRKY75, WRKY45, SHN1, and At2G20350, to contribute in the defense response against *Plasmodiophora brassicaegan* infection [37]. Moreover, knockdown of *TaEIL1* attenuated the growth of wheat stripe rust fungus (*Puccinias triiformis* f. sp. *tritici*) and enhanced wheat resistance to stripe rust [38]. In this study, *CsEIL2/4/5/8/10* showed significant upregulation at 2, 4, 6, and 8 dpi in response to *Xcc* infection, while the expression of *CsEIL1/7/9* was suppressed. In contrast, following *C*Las infection, *CsEIL2/4/5/8/10* exhibited significant downregulation, with only *CsEIL9* showing significant upregulation, suggesting that these genes may display conserved functions in sweet orange disease resistance through positive or negative regulation.

Overall, in this work we observed that *CsEIL* genes responded to different abiotic and biotic stresses to varying degrees. Based on our findings, it is posited that these genes play a dynamic regulatory role within the stress-induced gene regulatory network in sweet oranges. Therefore, further studies focusing on the functions and molecular mechanisms of *CsEIL* genes in the disease resistance of sweet oranges would deepen our understanding of their contribution to this process and allow for the potential development of novel tools and methods to enhance the resistance of this commercially important plant family to biotic and non-biotic stress.

## 4. Materials and Methods

### 4.1. Plant Materials, Hormone Treatment and Abiotic Stress

The plant materials were provided by the Changsha Branch of the National Citrus Improvement Center of Hunan Agricultural University. Plant materials of “Juxianglong” ice sugar orange were transplanted into the greenhouses using Danish peat soil and water-soluble fertilizers. The greenhouse temperature was maintained at 25 °C with a light period of 16 h and a dark period of 8 h.

Sweet orange seeds were meticulously disinfected and subsequently sowed in MS solid medium. They were then incubated in a plant incubator at 28 °C for 30 days, with a light period lasting 16 h and a dark period of 8 h. Well-developed seedlings were then transferred to Hoagland medium for a 2-day pre-culture period before undergoing stress treatments. For high-salt stress, the seedlings were exposed to Hoagland medium supplemented with 250 mol/L NaCl before being incubated at 28 °C with a photoperiod of 16 h light and 8 h darkness. For high- and low-temperature stresses, the seedlings were incubated at 42 °C and 4 °C with a photoperiod of 16 h light and 8 h dark, respectively. Each treatment group comprised 18 seedlings, and samples were collected at 0, 12, 24, and 48 hpt and stored at −80 °C until further use.

For hormone treatment, well-grown seedlings were transferred to a closed container of 100 mM ethylene glycol (ETH) solution, while the experimental control group was incubated with sterile distilled water at 25 °C. Samples were collected at 0, 1, 6, 12, and 24 hpt and kept at −80 °C until RNA extraction.

### 4.2. Xcc and CLas Immersions

The DL509 strain (Asian A line) of *Xcc* was isolated from the Changsha Branch of the National Citrus Improvement Center of Hunan Agricultural University and stored at −80 °C after purification. The DL509 strain was activated by streaking *Xcc* stock cultures on LB solid medium and incubating the plates at 28 °C for two days. Single colonies were selected and grown in 5 mL of liquid LB medium at 28 °C with shaking at 200 rpm for 18 h. Subsequently, 1 mL of bacterial solution was transferred into 50 mL of liquid LB medium and incubated at 28 °C with shaking at 200 rpm for 12 h. The bacterial cells were collected by centrifugation at 4 °C and 5000 rpm, resuspended in sterile water, and adjusted to an OD600 of 0.62 (approximately 10^9^ CFU/mL). A 10-fold dilution gradient was prepared, and the inoculation dose of *Xcc* was selected at a range of 10^4^–10^5^ CFU/mL. Subsequently, young leaves with consistent growth potential were selected for inoculation with *Xcc*.

The *C*Las materials were collected from diseased orchards in Hunan and Guangxi. Grafting transmission was performed following the method described by Zou et al. [39]. One-year-old healthy “Juxianglong” ice sugar orange with consistent maturity were selected as rootstocks. Leaf vein DNA was extracted every 3 months for CLas-positive detection using qRT-PCR. Three independent positive samples were collected and stored at −80 °C until RNA extraction.

### 4.3. RNA Extraction and qRT-PCR Analysis

Total RNA was extracted from sweet oranges using the TRIzol reagent (Ambion, Waltham, MA, USA) and purified using the EasyPure Plant RNA Kit (TianGen, Beijing, China). First-strand cDNA synthesis was performed using the NovoScript Plus All-in-One SuperMix kit (Novoprotein, Shanghai, China). The specific primers used for qRT-PCR are listed in Appendix A. Samples of the stems, leaves, and fruits were collected at the same time intervals [40]. RT-qPCR was performed on a CFX 96 qPCR instrument (Bio-Rad, Hercules, CA, USA) using a TransStart Tip Green qPCR SuperMix kit (TransGen, Beijing, China). The amplification of *CsCOX* was used as a reference control.

### 4.4. Bioinformatic Analysis

Information and sequences of *CsEIL* family genes were obtained from the Plant Transcription Factor Database v5.0 (http://planttfdb.cbi.pku.edu.cn/family. accessed on 14 October 2022). Public information on the sweet orange genome sequence and gene annotation was obtained from Phytozome v13 (https://phytozome-next.jgi.doe.gov. accessed on 14 October 2022) and the sweet orange gene from Huazhong Agricultural University (http://citrus.hzau.edu.cn/download.php. accessed on 14 October 2022). Multiple sequence alignments of the amino acid sequences of EIL members were created using MEGA11 version 11.0.11 with default parameters, and phylogenetic trees were drawn using the maximum-likelihood-based algorithm in MEGA11 software version 11.0.11 [41].

The chromosome locations of the *CsEIL* genes were extracted from the sweet orange genome annotation information and visualized using the standalone TBtools version 1.120 software package. Chromosome size and gene density were determined based on the sweet orange genome annotation information. The *CsEIL* gene replication events were analyzed using the Multiplex Tandem Scan Toolkit (MCScanX) program with the default settings. Sweet orange, Arabidopsis, apple, and poplar genome sequence files and gene structure annotation files were entered into One-Step MCScanX for covariance analysis and then visualized using the Dual Systeny Plot plug-in embedded in the TBtools version 1.120 software [42].

The theoretical isoelectric point (*p*I) and molecular weight (MW) of the protein sequences were calculated using the Protein Analysis System 3.0 (https://web.expasy.org/protparam. accessed on 15 October 2022) [43]. The subcellular localization signal of CsEIL was predicted using PlantmPLoc (http://www.csbiosjtu.edu.cn/bioinf/plant-multi. accessed on 15 October 2022). The gene structure of the EIL was obtained using the Gene Structure Display Server (http://lgsds.cbi.pku.edu.cn. accessed on 15 October 2022) based on the genomic and coding sequences. Conserved motifs of CsEIL were identified using the online MEME Suite program in the classical mode. Conserved structural domains were analyzed using the default mode on the SMART server (http://smart.embl.de. accessed on 15 October 2022).

To identify CREs in the promoter sequence of sweet orange *EIL* genes, a 2000 bp region upstream of the promoter was extracted and analyzed for CRE prediction using PlantCare (http://bioinformatics.psb.ugent.be/webtools/plantcare/html. accessed on 18 October 2022) [44], and then visualized using the Simple BioSequence Viewer plug-in embedded in the TBtools version 1.120 software for visualization and analysis. The CRE analysis statistics were produced using the CHIPLOT server (http://www.chiplot.online. accessed on 20 October 2022).

## 5. Conclusions

In this study, a comprehensive and systematic analysis of the EIL family of proteins was conducted in sweet orange, with a particular focus on their responses to canker and HLB infection. Ten *CsEIL* genes were identified and classified into three subfamilies based on their conserved motifs, which were unevenly distributed on the four chromosomes. The chromosomal position, gene replication, structural characteristics, conserved domains, motifs, CREs, and expression profiles (tissue specificity, response to citrus canker, and abiotic stress) of the *CsEIL* genes were analyzed. *CsEIL* genes are widely expressed in different tissues in response to high temperature, low temperature, and salt treatment, and are induced by ET. Notably, eight *CsEIL* genes were responsive to canker infection, nine *CsEIL* genes were responsive to HLB infection, suggesting their potential involvement in the immune defense response of sweet oranges. These findings provide a theoretical basis for further exploration of the role of the EIL family in resistance and susceptibility to canker and HLB disease and offer new insights for novel disease-resistant plant breeding approaches.

## Figures and Tables

**Figure 1 ijms-24-12644-f001:**
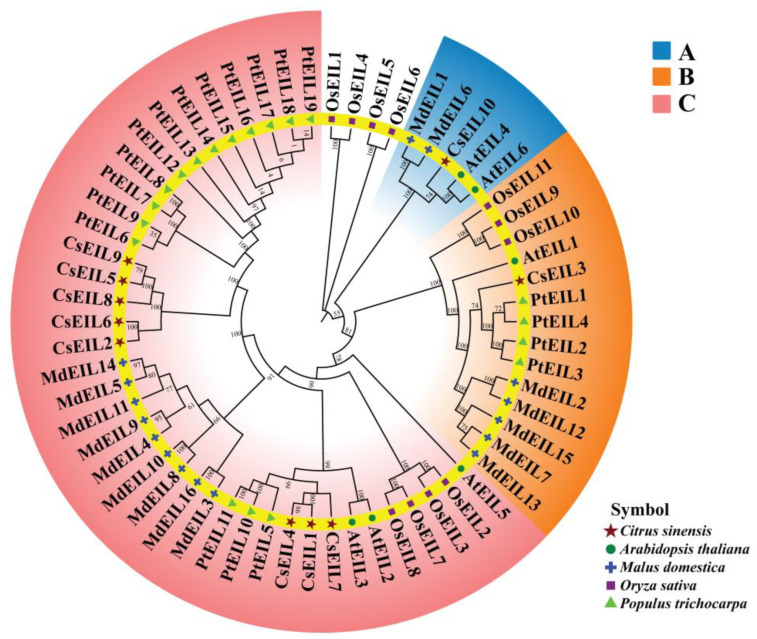
Phylogenetic analysis of EIL protein in Arabidopsis, poplar, rice, apple and sweet orange. Phylogenetic relationships of EIL proteins from *Citrus sinensis* (10), *Oryza sativa* (11), *Arabidopsis thaliana* (6), *Malus domestica* (16), and *Populus trichocarpa* (19) were inferred using the maximum likelihood method and MEGA v11 with default parameters. The EIL proteins were classified into three major groups: A, B, and C. Members of sweet orange, rice, Arabidopsis, apple, and poplar are represented by orange pentagrams, purple diamonds, blue circles, green squares, and orange triangles, respectively. The *EIL* gene IDs from sweet orange, rice, Arabidopsis, apple, and poplar are listed in Appendix A.

**Figure 2 ijms-24-12644-f002:**
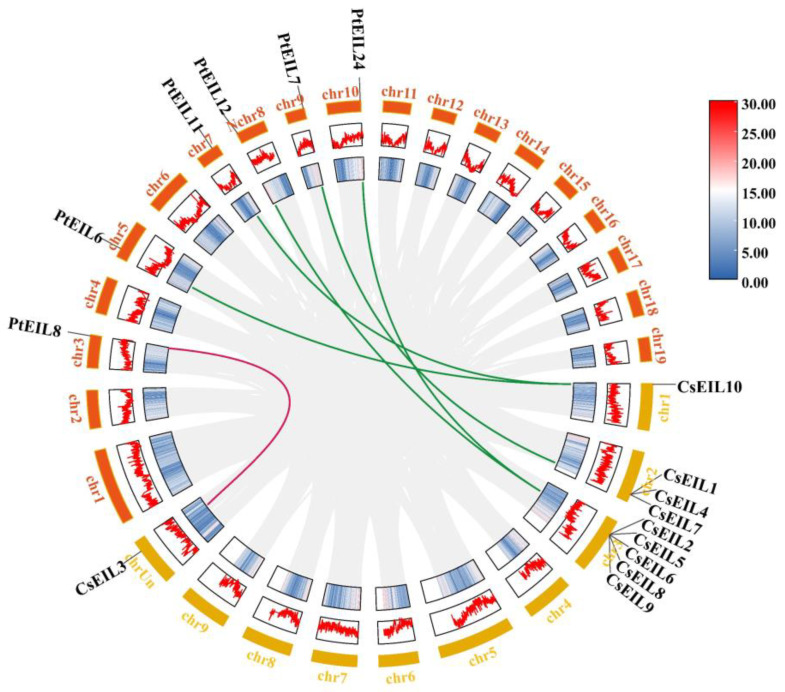
Duplication and collinearity analysis of EIL genes between sweet orange and poplar. The gray lines represent all collinear regions, and the other colored lines represent homologous gene pairs (the blue to red scale indicates the gene density from low to high).

**Figure 3 ijms-24-12644-f003:**
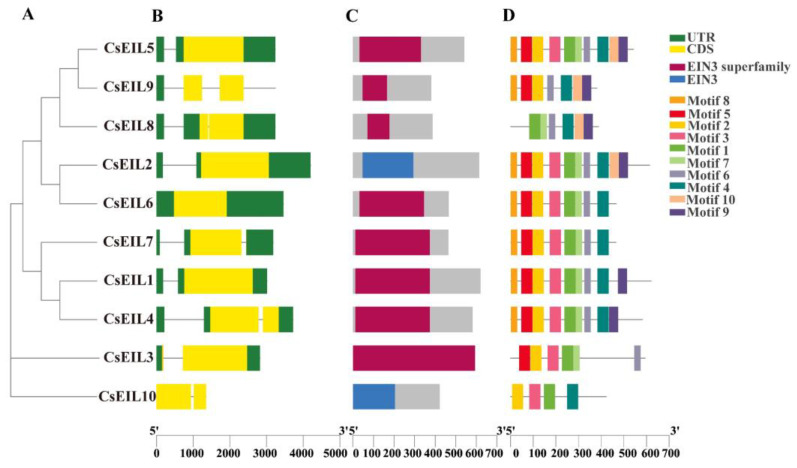
Phylogenetic analysis, protein domain and motif analysis of the CsEIL family. (**A**) Maximum likelihood phylogenetic tree of CsEIL amino acid sequences created using MEGA v11 software. (**B**) Gene structure of *CsEIL* in sweet orange, with CDS, UTR, and introns represented by yellow rectangles, green rectangles, and black lines, respectively. (**C**) Composition of the protein domains of CsEIL in sweet orange. (**D**) Conserved motifs of the CsEIL protein in sweet orange. The bottom scale is included to estimate the size of the protein and gene structures.

**Figure 4 ijms-24-12644-f004:**
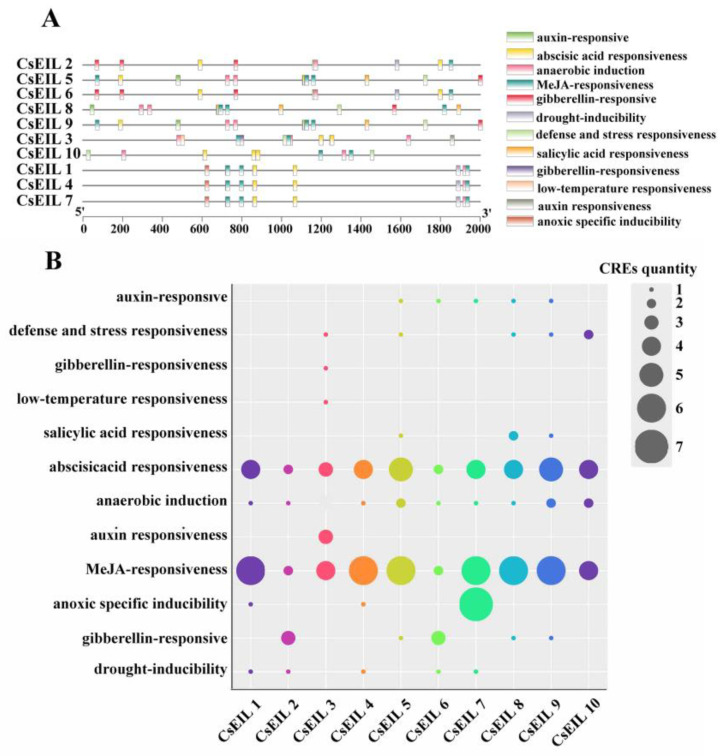
Analysis of *cis*-regulatory elements in the *CsEIL* gene promoter sequence. (**A**) Diagram outlining the various CREs located in the promoter sequence of the *CsEIL* genes. Each CRE is represented by a different colored box, and the 12 different CREs are described by their respective colors on the right side of the diagram. (**B**) More detailed analysis of the CREs, showing bubbles of varying sizes that represent the number of CREs present on the *CsEIL* genes. The bubbles are arranged from top to bottom to represent the different types of CREs, including auxin (IAA), defense and stress, gibberellins (GAs), low temperature, salicylic acid (SA), abscisic acid (ABA), anaerobic, auxin response, jasmonic acid (JA), hypoxia, and drought. Each bubble is colored differently to represent the different CREs present on the *CsEIL* genes.

**Figure 5 ijms-24-12644-f005:**
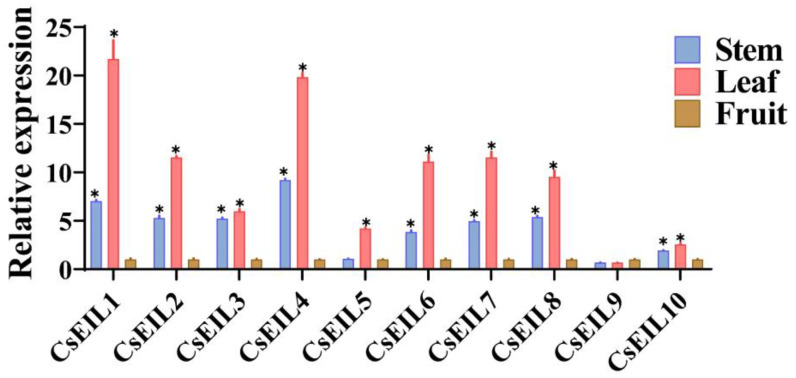
qRT-PCR identification of the *CsEIL* genes in different tissues of sweet orange. The experiment was conducted with three biological replicates. (*) denotes significant difference (*p* < 0.05).

**Figure 6 ijms-24-12644-f006:**
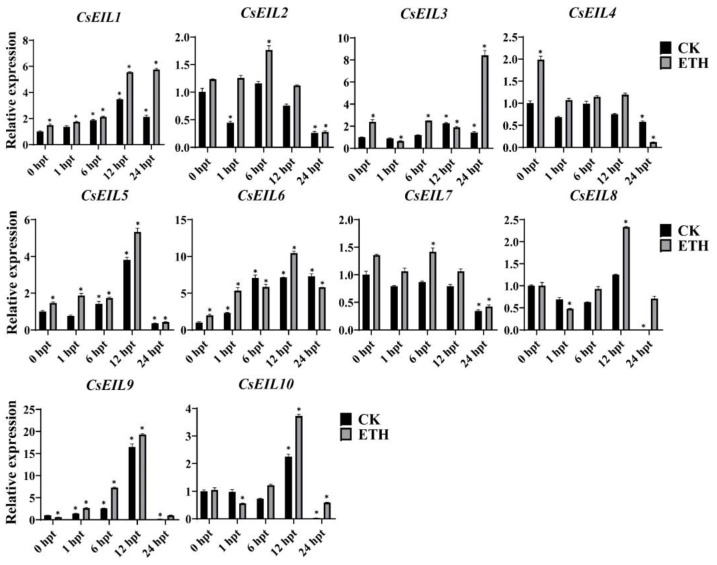
The expression levels of *CsEIL* genes under ET treatment. The gray bars represent the ET treatment, while the black bars represent the control. The experiment was conducted with three biological replicates, and (*) indicates a significant difference (*p* < 0.05).

**Figure 7 ijms-24-12644-f007:**
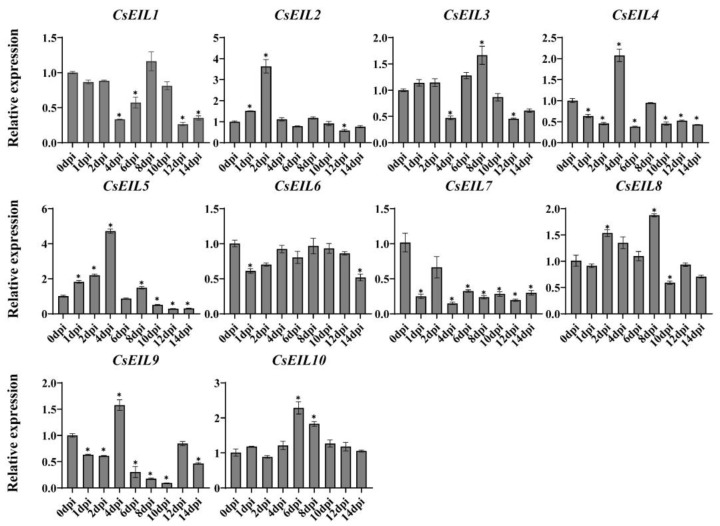
Expression levels of *CsEIL* genes after *Xanthomonas citri* inoculation. The X-axis represents different time points after inoculation treatment, while the Y-axis presents the relative gene expression levels of *CsEIL*. The experiment was conducted with three biological replicates in all panels, and similar outcomes were observed. The asterisk (*) indicates a significant difference (*p* < 0.05).

**Figure 8 ijms-24-12644-f008:**
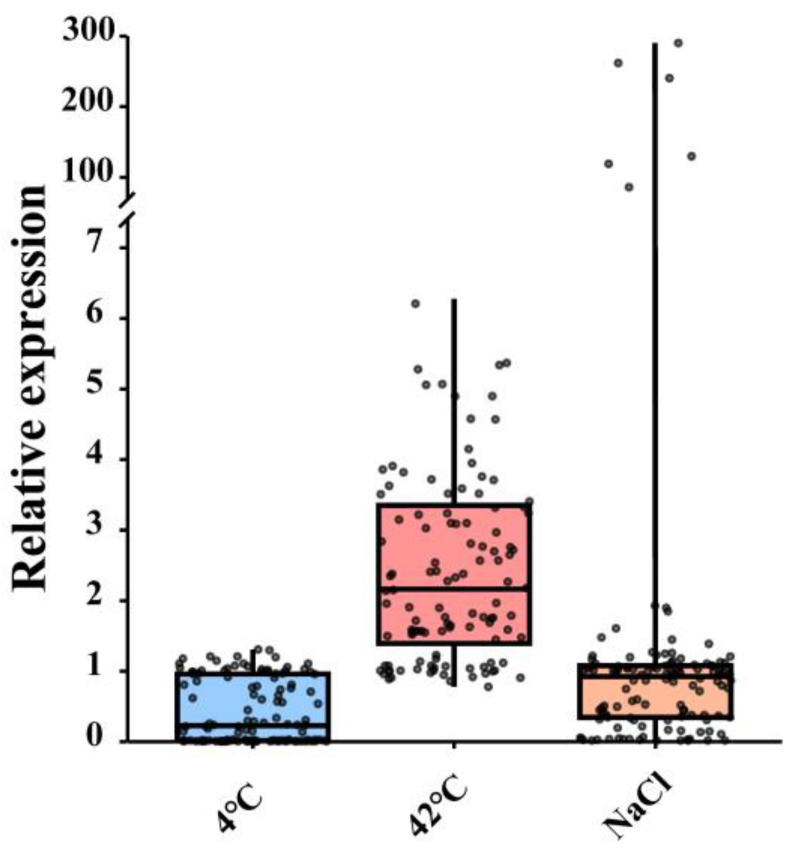
Expression levels of *CsEIL* genes under different abiotic stresses. The X-axis depicts various abiotic stressors, while the Y-axis presents the relative gene expression levels. The whiskers indicate the maximum and minimum values, the dots represent different levels of gene expression, the box displays the first and third quartile ranges, and the horizontal line represents the median expression level.

## Data Availability

Data is contained within the article or Appendix A.

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
