# Peer review of "Unleashing the Potential of EIL Transcription Factors in Enhancing Sweet Orange Resistance to Bacterial Pathologies: Genome-Wide Identification and Expression Profiling"

_ijms, 2023, doi:10.3390/ijms241612644_

Round 1

Reviewer 1 Report

In this study, the authors conducted a comprehensive and systematic analysis of ten EIL TF genes in sweet orange. Using quantitative reverse transcription polymerase chain reaction (qRT-PCR), the authors examined the expression of these ten genes in response to abiotic and biotic stresses, as well as in different tissues. As a result, they found that some EIL genes might be involved in infections caused by two pathogens, resulting in canker and Huanglongbing (HLB) diseases. Overall, I believe the purposes and results are suitable for publication. However, there are some points that need to be addressed before publication. Therefore, I suggest this manuscript undergo minor revisions before considering it for publication. My specific comments are as follows:

Title:

"Unleashing the Potential of EIL Transcription Factors in Enhancing Sweet Orange Resistance to Bacterial Pathologies: Genome-wide Identification and Expression Profiling"

Line 20-21:

"Putative cis-acting regulatory elements (CREs) associated with CsEIL were found to be involved in plant development, as well as responses to biotic and abiotic stress."

Line 21:

Please replace "qPCR" with "quantitative reverse transcription polymerase chain reaction (qRT-PCR)."

Line 27:

"Citrus Huanglongbing (HLB)"

Line 37:

"The regulatory network [1]." Please ensure there is a space between the sentence and the reference. Please check thoroughly throughout the manuscript for this error.

Line 47:

What is the full name of EIN3?

Line 48:

Please verify whether EIL1, representing EIN3-Like1, is correct.

Line 100:

Please regenerate the phylogenetic tree in Figure 1 using the maximum likelihood method instead of the neighbor-joining method with bootstrap replicates. Additionally, the quality of Figure 1 should be improved, particularly the font sizes, which are currently small.

The quality of Figure 3 should also be improved, as the characters are not readable. Please regenerate the phylogenetic tree in Figure 3 using the maximum likelihood method with bootstrap replicates.

Similarly, the quality of Figure 4 should be improved, as the characters are not readable.

The size of the graphs and font sizes in Figures 5 and 6 should be increased. Additionally, the "P" indicating probability should be in italics and capitalized.

The graphs in Figure 7 are fine. However, it would be better to increase the size of the graphs.

In my opinion, it would be desirable to prepare a new figure summarizing all qRT-PCR results for the ten CsEIL genes under different conditions using a heatmap.

Line 326:

Revise the degree sign for -80 °C.

Line 367:

Please check the version of the MEGA program.

Line 376:

The isoelectric point (pI) "P" should be in italics.

Author Response

Comment 1 (Title): "Unleashing the Potential of EIL Transcription Factors in Enhancing Sweet Orange Resistance to Bacterial Pathologies: Genome-wide Identification and Expression Profiling"

Response: Thank you very much for your suggested modifications to the article title. After thorough discussion and comparison, all the authors unanimously agree that the title you have provided is more suitable for our paper. We have made the necessary changes and highlighted them in red (Title).

Comment 2 (Line 20-21): "Putative cis-acting regulatory elements (CREs) associated with CsEIL were found to be involved in plant development, as well as responses to biotic and abiotic stress."

Response: Thank you very much for your modifications to this sentence, which have made it more refined. We have made the necessary changes and highlighted them in red (Line 20-22).

Comment 3 (Line 21): Please replace "qPCR" with "quantitative reverse transcription polymerase chain reaction (qRT-PCR)."

Response: Thank you for your careful review. Based on your feedback, we have made the necessary corrections and highlighted them in red (Line 22).

Comment 4 (Line 27): "Citrus Huanglongbing (HLB)"

Response: Thank you very much for your careful review. We have added spaces before the brackets and highlighted them in red (Line 28).

Comment 5 (Line 37): "The regulatory network [1]." Please ensure there is a space between the sentence and the reference. Please check thoroughly throughout the manuscript for this error.

Response: Thank you for your professional review. We apologize for our carelessness. We have thoroughly checked and corrected the reference formatting throughout the entire article and highlighted the changes in red (Line 38, Line 39, Line 43, Line 45, Line 47, Line 49, Line 51, Line 54, Line 56, Line 59, Line 62, Line 65, Line 74, Line 75, Line 77, Line79, Line85, Line89, Line250, Line256, Line258, Line263, Line264, Line277, Line284 Line300, Line303, Line305, Line353, Line364, Line375, Line383, Line386, Line395).

Comment 6 (Line 47): “What is the full name of EIN3?”

Response: Thank you for your thorough review, and we sincerely apologize for the oversight. The full name of EIN3 was missing in its first appearance in the article. We have now added the full name and highlighted it in red (Line 50).

Comment 7 (Line 48):“Please verify whether EIL1, representing EIN3-Like1, is correct.”

Response: Thank you for the reminder. After double-checking, we confirmed that the citation "EIN3-Like1" indeed refers to "ethylene-insensitive3-like1 (EIL1)" in Arabidopsis, with accession number AF004213.1.

Comment 7(Line 100): “Please regenerate the phylogenetic tree in Figure 1 using the maximum likelihood method instead of the neighbor-joining method with bootstrap replicates. Additionally, the quality of Figure 1 should be improved, particularly the font sizes, which are currently small.”

Response: Thank you very much for your valuable suggestions. We have redrawn the phylogenetic tree using the maximum likelihood method and have maximized the font size in Figure 1 to provide a more comfortable reading experience for the readers (Line106-113, Line116-117, Line375).

“The quality of Figure 3 should also be improved, as the characters are not readable. Please regenerate the phylogenetic tree in Figure 3 using the maximum likelihood method with bootstrap replicates.”

Response: Thank you very much for your valuable suggestions. We sincerely appreciate your feedback and will make efforts to improve the quality of Figure 3 to ensure better readability of the characters. We have already regenerated the phylogenetic tree in Figure 3 using the maximum likelihood method with bootstrap replicates (Line150, Line152).

“Similarly, the quality of Figure 4 should be improved, as the characters are not readable.”

Response: Thank you very much for your guidance. Regarding Figure 4, we will make efforts to enhance the image quality to ensure the readability of characters. We will reprocess the image and ensure that the display of Figure 4 is improved in the final submission version(Line170-171).

“The size of the graphs and font sizes in Figures 5 and 6 should be increased. Additionally, the "P" indicating probability should be in italics and capitalized.”

Response: Thank you very much for your valuable suggestions. We sincerely appreciate your attention to the details of Figures 5 and 6. We will make necessary adjustments by increasing the size of the graphics and fonts to improve readability. Additionally, we thoroughly reviewed the entire manuscript to ensure that the "P" indicating probability is presented in italics and capitalized, and the changes have been highlighted in red(line194, Line208, Line228, Supplementary figures 4), as you suggested. Your feedback will undoubtedly enhance the presentation of our research findings.

“The graphs in Figure 7 are fine. However, it would be better to increase the size of the graphs.”

Response: Thank you very much for your feedback. We appreciate your positive evaluation of the graphs in Figure 7. We have made efforts to increase the size of the graphs to enhance their visibility and clarity. Your valuable input is crucial for improving the presentation of our research findings (Line224).

“In my opinion, it would be desirable to prepare a new figure summarizing all qRT-PCR results for the ten CsEIL genes under different conditions using a heatmap.”

Response: Thank you very much for your feedback. We highly value your input, and your suggestion to use a heatmap to comprehensively present the qRT-PCR results for all ten CsEIL genes under different conditions is indeed a great idea. We have previously attempted to present the data in this way, but due to the involvement of multiple time points in the ethylene treatment and Xcc treatment, the resulting data became quite complex and could not be clearly represented in a single heatmap. Nonetheless, we will carefully consider your suggestion and explore if there are other methods to better visualize this data. In the meantime, we will continue to use the previous figures. Once again, we appreciate your thoughtful advice and attention. If you have any other questions or suggestions, we would be more than happy to listen. Thank you!

Comment 8(Line 326): “Revise the degree sign for -80 °C.”

Response: Thank you very much for pointing that out. We sincerely apologize for our oversight. We will review the entire article to ensure the correction of the degree sign and present it accurately in the revised manuscript. The revised areas have been marked in red(line324, Line331, Line333, Line335 Line338, Line339 ,Line344-357).

Comment 9(Line 367): “Please check the version of the MEGA program.”

Response: Thank you very much for pointing that out. We apologize for our oversight. We have checked the version of the MEGA program, and in the revised manuscript, it is accurately presented as MEGA11 version 11.0.11. The corrections have been highlighted in red(line374-375).

Comment 10(Line 376): “The isoelectric point (pI) "P" should be in italics.”

Response: Thank you for your suggestion. We apologize for the oversight. We have revised the manuscript, and now the isoelectric point (pI) "P" is presented in italics. The correction has been highlighted in red (line384, Supplementary Table S1).

Reviewer 2 Report

The manuscript "Unleashing the potential of EIL transcription factors in sweet orange resistance to bacterial pathologies: genome-wide identification and expression profiling" aims to apply a comprehensive genome-wide analysis to identify some sweet orange EIL TFs (Transcription Factors). Subsequently, to investigate various aspects of these TFs, including their chromosomal location, gene duplication, structural features, conserved structural domains, motifs, and cis-regulatory elements (CRE). Also, the study explored the expression patterns of the CsEIL genes in response to different biotic and abiotic stressors. The results obtained from this research serve as a valuable starting point for deeper investigations into the functional and mechanistic properties of EIL genes in sweet orange. This knowledge can contribute to a better understanding of their roles in the plant's response to stress and other biological processes. While the topic is of significant relevance and general interest to the journal's readership, several concerns must be addressed before publication.

·         The authors are highly recommended to avoid using a personal pronoun (e.g., We, our, etc.); they can use the third party in the past tense's passive voice.

·         The authors are strongly advised to carefully review the manuscript to address grammar and other editing issues, especially spacing between sentence and citation brackets.

·         To ensure reader comprehension, providing the full name associated with any abbreviation at its first mention in the manuscript is essential. This practice enables readers who may not be familiar with the abbreviated terminology to follow along and understand the content—for example, Lines 38 and 39. Also, provide full names for any abbreviation in the figures caption.

·         In the Material and Methods section, it is important to include or complete the sources of all chemicals, software, and equipment by adding the city, state, and country information. This additional detail provides readers with specific information about where these items were sourced, ensuring transparency, and facilitating reproducibility.

·         In the Material and Methods section, it is crucial to either provide a proper citation for the previously published method or offer comprehensive details about the methodology to ensure ease of reproducibility.

·         In Line 325, the authors omitted to mention crucial details about the high and low-temperature treatments, such as the duration of exposure. Additionally, they did not specify the light intensity used in the experiments. These omissions hinder a comprehensive understanding of the experimental conditions and may impact the reproducibility of the study.

·         Line 339, replace ‘r/min’ with ‘rpm’.

·         Supplementary Table S1, please make it one table in landscape orientation.

·         The resolution of all figures is currently very low, making them difficult to follow. To address this issue, please enlarge all figures and provide high-resolution versions of each.

·         In line 125, Un is not clear.

·         Upon reviewing lines 181 to 187 and examining Figure 5, there appears to be a discrepancy. The statement made by the authors, "The CsEIL9 was significantly expressed at high levels in all tissues and was particularly enriched in fruits compared to the other examined CsEIL genes," does not seem to align with the data presented in Figure 5.

Moderate editing of the English language is required.

Author Response

Comment 1: “The authors are highly recommended to avoid using a personal pronoun (e.g., We, our, etc.); they can use the third party in the past tense's passive voice.”

Response:Thank you very much for providing valuable suggestions. We sincerely appreciate your feedback and apologize for any oversight on our part. The authors have taken your suggestions into serious consideration and made necessary revisions. In the revised manuscript, personal pronouns have been avoided, and the third person passive voice in the past tense has been adopted. (line18-19, line93-94, line97, line102-103, line123-124, line130-142, line145-147, line159-160, line162-163, line165-166, line183-184, line217-219, line237-239, line277-281, line312-313, line343-344).

Comment 2: “The authors are strongly advised to carefully review the manuscript to address grammar and other editing issues, especially spacing between sentence and citation brackets.”

Response:Thank you very much for your valuable feedback. We highly appreciate your input and have carefully reviewed the manuscript to address grammar and other editing issues, especially spacing between sentences and citation brackets (Line 38, Line 39, Line 43, Line 45, Line 47, Line 49, Line 51, Line 54, Line 56, Line 59, Line 62, Line 65, Line 74, Line 75, Line 77, Line79, Line85, Line89, Line250, Line256, Line258, Line263, Line264, Line277, Line284 Line300, Line303, Line305, Line353, Line364, Line375, Line383, Line386, Line395).

Comment 3: “To ensure reader comprehension, providing the full name associated with any abbreviation at its first mention in the manuscript is essential. This practice enables readers who may not be familiar with the abbreviated terminology to follow along and understand the content—for example, Lines 38 and 39. Also, provide full names for any abbreviation in the figures caption.”

Response: Thank you for your valuable advice. We truly appreciate your feedback and have taken it into serious consideration. To enhance reader comprehension, we will make sure to provide the full name associated with any abbreviation at its first mention in the manuscript. This will help readers who may not be familiar with the abbreviated terminology to follow and understand the content more easily, as mentioned in Lines 38 and 39. Additionally, we will include full names for any abbreviations in the figures' captions as well. Your guidance will undoubtedly improve the clarity and accessibility of our manuscript. (line39-41, line158, line172, line209-210, line225 and Supplementary figures 3).

Comment 4: “In the Material and Methods section, it is important to include or complete the sources of all chemicals, software, and equipment by adding the city, state, and country information. This additional detail provides readers with specific information about where these items were sourced, ensuring transparency, and facilitating reproducibility.”

Response: Thank you for your valuable feedback. We appreciate your suggestion, and we will make sure to include or complete the sources of all chemicals, software, and equipment in the Material and Methods section by adding the city, state, and country information. This additional detail will provide readers with specific information about the origins of these items, ensuring transparency, and facilitating reproducibility of our study (line321-322).

Comment 5: “In the Material and Methods section, it is crucial to either provide a proper citation for the previously published method or offer comprehensive details about the methodology to ensure ease of reproducibility.”

Response: Thank you for your valuable feedback. We understand the importance of providing proper citations or detailed methodology in the Material and Methods section for reproducibility purposes. In our revised manuscript, we will ensure to either cite the previously published method appropriately or provide comprehensive details about the methodology used (line326-335).

Comment 6: “In Line 325, the authors omitted to mention crucial details about the high and low-temperature treatments, such as the duration of exposure. Additionally, they did not specify the light intensity used in the experiments. These omissions hinder a comprehensive understanding of the experimental conditions and may impact the reproducibility of the study.”

Response: Thank you for bringing these omissions to our attention. We apologize for the oversight. In our revised manuscript, we will include the crucial details about the high and low-temperature treatments, including the duration of exposure(line326-335).

Comment 7: “Line 339, replace ‘r/min’ with ‘rpm’.”

Response: Thank you for pointing that out. We have made the necessary correction, and 'r/min' has been replaced with 'rpm' throughout the manuscript (line345, line347, line348,).

Comment 8: “Supplementary Table S1, please make it one table in landscape orientation.”

Response: Thank you for your suggestion. We have made revisions to Supplementary Table S1. Since a horizontal table cannot fully display all the data, we have now presented it as a single vertical table to enhance readability (Supplementary Table S1).

Comment 9: “The resolution of all figures is currently very low, making them difficult to follow. To address this issue, please enlarge all figures and provide high-resolution versions of each.”

Response: Thank you for bringing this matter to our attention. We will enlarge all the figures and fonts in the figures and ensure that high-resolution versions of each figure are provided in the revised manuscript to improve the clarity and readability of the figures (line113,line135, line150, line170-171, line192, line205, line224, line242, Supplementary figures 1, Supplementary figures 2, Supplementary figures3, Supplementary figures4).

Comment 10: “In line 125, Un is not clear.”

Response: Thank you for the clarification. In line 125, "Un" refers to "chromosome Un," which indicates an unplaced or unlocalized region in the genome assembly. We have now added an explanation for "chrUn" in the revised manuscript to avoid any confusion for the readers (line126-127).

Comment 11: “Upon reviewing lines 181 to 187 and examining Figure 5, there appears to be a discrepancy. The statement made by the authors, "The CsEIL9 was significantly expressed at high levels in all tissues and was particularly enriched in fruits compared to the other examined CsEIL genes," does not seem to align with the data presented in Figure 5.”

Response: Thank you for your suggestion. We have carefully reviewed lines 181 to 187 and Figure 5. We acknowledge the discrepancy between the statement and the data presented in Figure 5. The expression pattern of CsEIL9 in Figure 5 does not support the claim that it is significantly expressed at high levels in all tissues and particularly enriched in fruits compared to other CsEIL genes.

Upon further investigation, we found an error in the interpretation of the data. The correct expression pattern should be stated as follows: Upon further investigation, we found an error in the interpretation of the data. The correct expression pattern should be stated as follows: CsEIL1/2/4/6/7/8 displayed significant expression level differences among all tissues, with expression levels in leaves being higher than in stems. We have made the necessary revisions to the manuscript to reflect this correction (line183-188, line277-281).

Moreover, this manuscript was proofread by a native English professional with science background.

Round 2

Reviewer 2 Report

Thanks, the authors responded to all of comments!

Minor editing of English language required